# Displacement Estimation via 3D-Printed RFID Sensors for Structural Health Monitoring: Leveraging Machine Learning and Photoluminescence to Overcome Data Gaps

**DOI:** 10.3390/s24041233

**Published:** 2024-02-15

**Authors:** Metin Pekgor, Reza Arablouei, Mostafa Nikzad, Syed Masood

**Affiliations:** 1Department of Mechanical and Product Design Engineering, Swinburne University of Technology, Hawthorn, VIC 3122, Australia; mnikzad@swin.edu.au (M.N.); smasood@swin.edu.au (S.M.); 2Data61, Commonwealth Scientific and Industrial Research Organisation, Pullenvale, QLD 4069, Australia; reza.arablouei@csiro.au

**Keywords:** RFID, sensor, additive manufacturing, structural health monitoring, direction of arrival estimation, machine learning, photoluminescence

## Abstract

Monitoring object displacement is critical for structural health monitoring (SHM). Radio frequency identification (RFID) sensors can be used for this purpose. Using more sensors enhances displacement estimation accuracy, especially when it is realized through the use of machine learning (ML) algorithms for predicting the direction of arrival of the associated signals. Our research shows that ML algorithms, in conjunction with adequate RFID passive sensor data, can precisely evaluate azimuth angles. However, increasing the number of sensors can lead to gaps in the data, which typical numerical methods such as interpolation and imputation may not fully resolve. To overcome this challenge, we propose enhancing the sensitivity of 3D-printed passive RFID sensor arrays using a novel photoluminescence-based RF signal enhancement technique. This can boost received RF signal levels by 2 dB to 8 dB, depending on the propagation mode (near-field or far-field). Hence, it effectively mitigates the issue of missing data without necessitating changes in transmit power levels or the number of sensors. This approach, which enables remote shaping of radiation patterns via light, can herald new prospects in the development of smart antennas for various applications apart from SHM, such as biomedicine and aerospace.

## 1. Introduction

Structural health monitoring (SHM) involves gathering essential data from structures using a multitude of sensors. With the increasing complexity of sensor systems, effective and efficient management and interpretation of data is crucial for informed decision-making. Data analysis in SHM aims to convert sensor data into actionable insights and broader systemic understanding. There are two primary approaches for analysis in engineering structures: physics-based (or model-based) and data-driven methods [1]. Physics-based approaches involve the development of first-principle models, such as finite-element analysis, which map the physical characteristics of a system. These models are then juxtaposed with sensor data to assess the condition of a structure. On the other hand, data-driven methods involve analyzing data directly obtained from the structure, often using machine learning (ML) techniques. They typically use data from both healthy and defective structures to develop models capable of recognizing patterns and distinguishing various structural health states. Data-driven methods are increasingly preferred over physics-based ones due to their effectiveness and adaptability in handling complex, high-dimensional datasets without the need for explicit physical models. These methods leverage computational algorithms to identify patterns and insights directly from the data, offering more flexible and potentially more accurate predictions, especially in systems where the underlying physics are poorly understood or too complex to model accurately [1]. Most supervised ML methods are effective even with relatively small datasets [2].

Common SHM techniques encompass visual and camera-based inspections, typically employed as part of non-destructive testing (NDT) methodologies, frequently accompanied by ML algorithms. Image-recognition-based approaches generally necessitate collection of extensive data, a considerable portion of which may be superfluous, such as redundant pixel information or extraneous background elements, largely attributable to environmental variables [3].

The use of RFID sensors in SHM has spurred various new ML applications. These include moisture detection using artificial neural networks (ANNs) [4], corrosion detection through principal component analysis (PCA) [5], crack detection via PCA [6], and people activity recognition with convolutional neural networks (CNN) and deep neural networks (DNN) [3]. RFID systems coupled with ML typically involve multiple sensors/antennas and sophisticated RFID systems. RFID sensors can be either surface-mounted or embedded within the work environment. They are widely utilized in SHM systems for monitoring various parameters such as temperature, cracks, strain, corrosion, pH levels, and displacement. However, there still are outstanding challenges such as effective embedding and signal attenuation, especially in indoor settings [7]. Compliance with RFID standards can alleviate many operational challenges, and the precision of the system can be significantly improved through various measures. Enhancing system accuracy can be achieved by augmenting the array of antennas and sensors, both physical and virtual. Additionally, increasing the sampling frequency and utilizing higher frequency bands, such as millimeter-wave signals, contribute to system refinement. The adoption of robust techniques, including fingerprinting and hybrid methods that integrate time difference of arrival (TDOA), time of arrival (TOA), and direction of arrival (DOA) with the received signal strength indicator (RSSI), can further enhance performance. The strategic deployment of these methods across sensor platforms ensures a more accurate and reliable RFID system.

RF signals encounter a range of transmission challenges. These challenges include multipath effects, line of sight and non-line of sight complications, interferences, varying ambient RF energy levels, malfunctions in sensor arrays, issues with sensor orientation, and diverse effects induced by antennas or the environment. Such difficulties arise due to the inherent properties of RF signals, which include reflection, refraction, scattering, and absorption within the medium they traverse.

Antennas play a pivotal role in RF systems, as they are the primary medium for signal transmission and reception. While the fundamental principles of operation for antennas in transmitting and receiving signals are similar, the key distinction lies in the direction of signal flow. The effectiveness and quality of communication through an antenna are largely dependent on its radiation pattern. Beamforming technology, which is utilized in both transmitter and receiver sensor arrays, significantly enhances signal reception from specific directions. This technology allows for spatially selective transmission and reception, a feature referred to as directivity. Directivity is determined based on azimuth angles in the horizontal plane and elevation angles in the vertical plane. The application of beamforming extends across various fields, including radar, sonar, seismology, wireless networking, radio astronomy, and the medical industry. It is also applicable to both sound and radio waves.

DOA estimation algorithms are broadly categorized into traditional, subspace-based, maximum likelihood, and ML-based methods [8]. Classical methods for DOA estimation involve calculating a spatial continuum and identifying DOAs by locating local maxima. However, they suffer from angular resolution limitations. As a result, higher-angular-resolution subspace methods like multiple signal classification (MUSIC) and estimation of signal parameters via rotational invariance techniques (ESPRIT) are more prevalent [9]. Traditional methods like the minimum variance distortionless response (MVDR) and delay and sum beamforming, along with subspace and maximum likelihood methods, are generally more complex and less accurate compared to ML-based algorithms. ML-based approaches offer unique advantages in complex scenarios over other approaches. They are also utilized for determining antenna element failures, antenna positioning, and radiation pattern issues during the design and application phases and for enhancing resolution and optimizing beamforming [10,11].

RFID systems, which include RFID readers, antennas, and tags, operate on two-way communication—the interrogation signal from the reader and the backscattering signal from the tags. Several factors on both the reader and tag sides significantly influence signal transmission. On the reader side, the position of antenna arrays, receiver sensitivity gain, amplifier gain, coding parameters like time delay, frequency, bandwidth, phase noise, as well as the quality of cables and connectors are crucial for accurate functioning. On the tag side, the positioning of sensor arrays and ambient energy levels are key considerations for reducing errors in DOA estimation.

In this research, building on previous studies, we estimate the direction of arrival (DOA) using ML algorithms with a 3D-printed passive ultra-high frequency (the radio frequency range between 300 MHz and 3 GHz) (UHF) RFID sensor array mounted on the structure’s surface. This approach enables accurate remote estimation of object displacement for SHM purposes. We initially conduct experiments with a few sensors. As the number of sensors increases, the issue of missing data arises. To address this challenge, we implement a novel approach that incorporates a photoluminescent material to enhance sensor sensitivity. This strategy leads to improved signal retention and a reduction in data loss, effectively countering the issue of missing readings often encountered in applications involving multiple RFID sensors/tags. This innovative solution not only addresses the missing data problem but also significantly enhances the accuracy of our system. Consequently, passive RFID tags are successfully utilized as displacement sensors in SHM, showcasing the practicality and effectiveness of this approach in real-world applications.

## 2. Review of RFID Positioning Systems

Ye et al. [12] provide a categorization of RFID positioning systems, taking into account various factors such as the environment, application, and sensor type. These systems are classified as indoor or outdoor, absolute or relative, active or passive, and ubiquitous. Ubiquitous positioning systems are distinguished by their high accuracy, real-time estimation capabilities, and versatility in handling complex dimensions, including 2D and 3D, making them suitable for both indoor and outdoor environments. The term ‘ubiquitous’ signifies their omnipresence, available at any time and place. As such, RFID-based damage detection systems in SHM are anticipated to be ubiquitous. While environmental factors like temperature, humidity, and multipath effects pose challenges, these can be addressed using ML algorithms [13]. However, it is equally important to ensure the self-protection of RFID sensors against physical degradation in SHM applications. The advancement of 3D-printable embedded RFID sensors caters to this need by providing robust physical protection. Additionally, this technology facilitates precise sensor calibration during the 3D design phase, enhancing the overall effectiveness and reliability of SHM systems.

Cheng et al. [14] provide a classification of RFID positioning based on dimensions. The one-dimensional (1D) positioning system is categorized into absolute and relative types. Absolute positioning identifies a specific target’s location, while relative positioning assesses spatial relationships between multiple targets. Although absolute positioning systems are prone to higher errors, relative positioning systems more easily determine objects’ coordinates and their sequence. Spatial–temporal phase profiling (STPP) [15], relative localization method of RFID tags via phase and RSSI based on deep learning (PRDL) [16], and human movement-based relative localization (HMRL) [17] are examples of 1D RFID positioning methods in the literature.

Two-dimensional (2D) positioning systems can be divided into ranging and non-ranging positioning methods [14]. Ranging methods utilize TOA, TDOA, DOA, or RSSI [18]. TOA- and TDOA-based methods are known for their complex calculations and the need for exact synchronization to accurately estimate locations, a requirement that is not necessary for the RSSI-based method. Although DOA-based methods offer high accuracy, they are susceptible to non-line-of-sight (NLOS) issues, which can be a significant limitation in complex indoor environments [18]. To overcome these challenges, Wang et al. [19] have developed a method that integrates the DOA- and RSSI-based techniques. In their method, a phased array antenna is used to measure the azimuth angle and signal strength of tags. These data are then processed through a neural network algorithm, which converts the signal strength into distance measurements, enabling precise target positioning. This approach effectively combines the strengths of DOA- and RSSI-based methods, enhancing the accuracy and applicability of 2D positioning systems, particularly in environments where traditional methods face limitations.

According to Cheng et al. [19], RFID positioning techniques are broadly categorized into propagation-based and fingerprint-based models. In the propagation-based method, distance is initially estimated using RSSI data, followed by the application of trilateration. However, this method tends to be less accurate when compared to the fingerprint method, which directly maps collected RSSI data to specific locations without the necessity of distance estimation. Location determination in the fingerprint method is more precise, achieved through various matching techniques. This method is particularly well-suited for RFID applications as it does not rely on knowing the precise locations of the reader and sensors. Additionally, multidimensional scaling (MDS)-based location algorithms, particularly in the millimeter-wave band, address the power and bandwidth constraints typically associated with UHF and ultra-wideband UWB systems as regulated by industry standards [20]. For indoor fingerprinting applications using RSSI, continuous wavelet transforms are employed to provide crucial time-frequency information, further improving the precision and reliability of RFID positioning in various environments.

Ranging 2D positioning methods, such as LANDMARC [21], VIRE [22], BVIRE [23], ANTSpin [16], and LF-based indoor localization systems [24], have various applications. LANDMARC, known for its accuracy ranging between 1.5 m to 2 m with a 50% probability, has faced challenges like high costs and limitations due to multipath effects in enclosed spaces [22]. However, significant improvements have been made to LANDMARC over time. VIRE brought advancements to LANDMARC by incorporating virtual tags and threshold filters, which significantly increased the accuracy from 17% to 73% [14]. Further developments included BVIRE, which took into account the boundaries of virtual tags, and the integration of the KNN algorithm with Taylor series expansion to enhance location estimation [22]. LANDMARC’s accuracy saw improvements of 38% and 55% with the introduction of weighted path loss (WPL) and extreme learning machine (ELM), respectively [25]. Changzhi et al. introduced a method that combines Gaussian filtering with ELM to correct signal fluctuations. This approach notably improves the accuracy and speed in locating RFID tags [13]. These evolving methods reflect the continuous advancements in RFID technology, particularly in enhancing the accuracy and reliability of RFID-based positioning systems.

Three-dimensional positioning methods, like APM, APAA, 3DinSAR, VLM, and TagSpin, provide comprehensive spatial location information [14]. ML algorithms such as logistic regression (LR), support vector machine (SVM), and decision tree (DT) classifiers are used to distinguish valid data from false positives [26,27]. Techniques to detect false positives include sliding windows, offline cleaning, extra hardware, RSSI, phase, and doppler-based methods. For instance, sliding windows use fixed-size windows based on tag occurrence and timestamps to eliminate noise and duplications. Offline cleaning employs a hidden Markov model to find missing detections [27]. Extra hardware involves using additional antennas and tags, while RSSI is commonly used to separate static and mobile tags. ML algorithms like logistic regression, multilayer perception (MLP), DT classifiers, and rule-based classifiers can work with RSSI data. Phase and doppler methods are helpful for moving tags, with k-means clustering evaluating phase variation and speed data. Algorithms like ANN, LR, SVM, DT, random forest, and XGBoost have been applied with RSSI for real-time false-positive estimation [27]. Additionally, subarray sampling via Monte Carlo simulations has been used instead of full antenna arrays [28]. In conclusion, model-based DOA estimation algorithms such as maximum likelihood, generalized least squares (GLS), and residual sum of squares (SSR) estimators are less accurate and more complex compared to data-driven ANN algorithms.

RFID systems exhibit unique yet consistent characteristics in their scattering parameters and noise rates under similar measurement conditions. These characteristics can significantly influence the raw data collected and used in the estimation process. ML algorithms can learn these unique features, enabling them to make accurate approximations and reduce error. In research using the phase-comparison mono-pulse method, akin to a radar DOA technique, ML algorithms are employed to estimate the positions of UHF RFID tags [29]. Various algorithms, including J48 tree, MLP, partial decision tree (PART), SVM, naïve Bayes, expectation-maximization (EM) clustering, and k-means clustering, have been tested.

## 3. Materials and Methods

In this section, we outline the test procedures using both the classical method and the new encapsulation technique. The detailed implementation of 3D encapsulation of RFID sensors via additive manufacturing (AM), as well as a thorough discussion on related signal modeling, underlying assumptions, fundamental principles of RFID sensing, and the data processing pipeline, are elaborated in [30]. Here, the objective is to explore the development of new methods for displacement estimation in SHM utilizing 3D-printed RFID sensors. Our methodology involves the estimation of DOA values through the application of ML algorithms, embracing both supervised learning paradigms of classification and regression. Given the data-driven nature of our approach, which necessitates high-quality data, we introduce a novel method to enhance RFID signal levels through exploiting the synergistic impact of photoluminescence on RF signals and antenna radiation patterns. The proposed technique helps address potential data gaps and improve the reliability and accuracy of ML-based displacement estimation in SHM applications.

### 3.1. Test Procedure

We employ RFID technology to estimate positional variations (displacements) of tagged objects in a horizontal plane for SHM purposes. The hardware setup includes an ST25RU3993 RFID reader evaluation board, a linear polarized antenna, and ALN-9762-WRW meander-type passive tags. Tests are carried out in a room constructed with wood/concrete, metal cladding, and in a non-echoic RF dosimetry laboratory. RSS levels from RFID sensor arrays, arranged in a uniform linear array (ULA) formation, are used to estimate the DOA using ML algorithms.

### 3.2. Test Setup

The problem is investigated using a platform built with an RFID sensor array consisting of eight identical 3D-printed RFID tags encapsulated with 2.5 cm thick polypropylene (PP) on a wooden sliding door. The stationary RFID reader antenna, single-polarized with a gain of 6 dB_i_, is positioned 3.25 λ away from the sensor array. The RSSI-featured RFID reader collects all incident signal data, including signal strength and phase (Figure 1). These data are processed at a specific azimuth angle of the array to estimate the object’s position variation through ML algorithms.

Initially, an eight-element 3D-printed sensor array is mounted 75 cm above the ground at the center of the sliding door (Figure 2). The RSSI data, including the sensor’s electronic product code (EPC), signal amplitude, and I/Q values, are gathered while incrementally moving the sliding door by 1 cm steps. For every 1 cm displacement, an angle of 0.538 degrees is swept, and sensor data are measured from 105 cm away using an RFID reader (Figure 2). The collected data are then labeled with physical displacement and angle values, while parameters like frequency, bandwidth, gain, and distance between the reader antenna and sensor array are also recorded.

### 3.3. Measurements

We conduct three distinct tests using a monostatic RFID reader to assess the impact of transmitter parameters on ML algorithms:Test-1—wideband: The RFID reader is set to a wide frequency range (902.750–927.250 MHz). The transmitter and receiver attenuators/amplifiers are adjusted automatically.Test-2—narrow-amplifier: The RFID reader is set to function in a narrow band (921 MHz), with the transmitter and receiver attenuators/amplifiers again left on the automatic mode.Test-3—narrow-constant: The RFID reader employs a narrowband signal with constant attenuator/amplifier levels (0 dBm) at both the transmitter and receiver.

The measurement setup for all tests includes:
angle change of 0.538°/cm for each measurement step (d=3.25λ≈105.86 cm for 921 MHz, $\tan^{-1}\left(1/105.86\right)$ ≈ 0.538°)measurement distance 3.25 λnumber of array elements n=8array element spacing d=0.25λnumber of incoming signals K≤8number of data samples k=1000 each 10 ssystem frequency f=902–927 MHz for Test-1, f= 921 MHz for Test-2 and Test-3system bandwidth 15 MHz for Test-1, 200 kHz for Test-2 and Test-3RFID reader receiver attenuation: variable for Test-1 and Test-2, 0 dB for Test-3 (sensitivity mode is OFF)RFID reader transmitter attenuation: variable for Test-1 and Test-2, 0 dB for Test-3 (AGC is OFF).

As the utilized RFID reader is monostatic, the transmitted signal is adjusted for receiver level sensitivity via gain control. Oscillator settings change the frequency. It is known that setting frequency, bandwidth, automatic gain control (AGC), and I/Q channel switching to automatic mode can result in unpredictable RSSI readings [4]. Additionally, RFID reader sensitivity is a limiting factor for the radiation range, and the frequency power sensitivity may vary based on RFID chip characteristics [31]. Considering these factors, Test-1 is conducted with a wideband setting, while Test-2 and Test-3 are carried out with a narrowband setting. In Test-1 and Test-2, the receiver and transmitter gain amplifiers are set to automatic modes, but for Test-3, they are fixed.

### 3.4. Machine Learning Approach

The learning process in ML, regardless of the algorithm type, involves several crucial stages. These include collecting and preparing data, selecting a suitable model, training the model using the available data, selecting a validation method, iteratively examining and adjusting the model for optimal fit, and finally, using the best-performing model to make predictions.

### 3.5. Data Collection and Preparation

Our objective is to estimate the structural displacement in terms of horizontal azimuth angle using a stationary RFID reader that measures the RSSI levels of signals backscattered from RFID sensors located on or inside the structure. Key system parameters include:
sensor labels $S_{1},S_{2},\cdots S_{n}$sensor RSSI mean values $R_{S1},R_{S2},\cdots R_{Sn}$sensor I values $I_{S1},I_{S2},\cdots I_{Sn}$sensor Q values $Q_{S1},Q_{S2},\cdots Q_{Sn}$ML model f($\theta_{pm}$)known azimuth angles (target outputs) $\theta_{1},\theta_{2},\cdots \theta_{m}$predicted azimuth angles $\theta_{p1},\theta_{p2},\cdots \theta_{pm}$

Data collected by the RFID reader are initially prepared for algorithmic processing. RSSI values in dBm are recorded in a CSV file. Null or invalid RSSI values are set to −65 dBm. Each sensor and sensor array is labeled, and the data are associated with an azimuth angle based on the sliding door’s position. The data collected by the RFID reader are sent to a local computing device for analysis. The method of data transmission and the communication channel used can differ based on the specific requirements of the application.

### 3.6. Enhancing the Model Performance

While ML strategies are powerful, they are not always a panacea for prediction challenges. The accuracy of an ML model is largely contingent on the quality and consistency of the training dataset used during its development. High precision is typically achieved in regions of the feature space that have been thoroughly covered in the training phase [32]. When working with ML algorithms, several strategies should be considered to achieve the highest accuracy or to enhance model performance:-Utilize more data: If there are no missing data, use a more extensive dataset rather than being limited to a smaller one.-Address outliers and missing data: identify outliers and missing data, treating them as important components of your dataset [33].-Feature engineering: employ feature engineering techniques using the existing dataset to create more informative features.-Feature selection: carefully select features that have the most significant impact or contribution to the variable being predicted.-Model selection: choose the appropriate model based on the nature of the data and the specific problem being addressed.-Parameter adjustment: fine-tune the model parameters to optimize performance.-Cross-validation: implement cross-validation techniques to assess the effectiveness of the model and to prevent overfitting.

### 3.7. Test Results for Ten Displacement Steps

Here, we use eight RFID sensors (n=8) and ten different displacement steps (m=10), corresponding to known azimuth angles $\theta_{1}$, $\theta_{2}$, …, $\theta_{m}=0.538,\;2\times 0.538,\;\ldots ,\;10\times 0.538{}^{\circ}$. The dataset encompassing these angles is used to train various classification and regression models. Out of 176 observation data points, 85% are allocated for training and 15% for testing and validation. The test results are presented in Figure 3 and Figure 4.

Data of Test-1 (wideband), Test-2 (narrowband with automatic amplifier gain), and Test-3 (narrowband with constant amplifier gain) are employed for training both classification and regression algorithms. Interestingly, the data collected under wideband conditions are the most effective for training. This dataset yields favorable results across numerous classification and regression algorithms. For all tests (Test-1, Test-2, and Test-3), the ensemble classification and Gaussian regression models emerged as the best-performing ones.

### 3.8. Test Results for Eighty Displacement Steps

The same tests are repeated for eighty displacement steps. As the number of steps increases, so does the distance, leading to weaker or even disappearing backscattering signals. Consequently, the incidence of missing data rises to 21%. This dataset, without altering parameters like the RFID reader’s location or amplifier gain, is then used to train various ML algorithms. The performance of the resulting classification and regression models is poor, around 20% accuracy at best.

To enhance algorithm performance, we employ several imputation and deletion techniques to address missing data. These techniques include using the mean, deviation, last observation carried forward, and listwise deletion methods [34]. Using these approaches, the ML performance only improves to a maximum of 62% accuracy, as shown in Figure 5. This highlights the challenge of handling missing data and their significant impact on the effectiveness of ML algorithms in practical applications.

### 3.9. Challenges and Novel Solutions for Missing Data in the RFID Systems

Missing data occur when labeled objects are not scanned despite their presence. This phenomenon is recognized in the literature as the second-most significant challenge in RFID deployment [35]. The primary causes of these irregularities include tag collisions, detuning, and misalignment. Furthermore, multipath propagation is a major factor in degrading the performance of RFID systems, resulting in missing readings [35]. To address this problem, both physical and numerical approaches are employed. These strategies aim to minimize missing data and improve the overall accuracy and reliability of RFID systems.

The RFID backscattering communication operates in two distinct steps. In the first step, known as the downlink, the reader sends an interrogation message, which is demodulated by the tag. The second step, the uplink, involves the tag transmitting a modulated signal back to the reader. Amplitude-shift keying (ASK) and phase-shift keying (PSK) are commonly used modulation types in RFID systems. Passive tags require activation by the reader’s radiation power and become active when their energy levels reach the chip’s threshold. The sensitivity of RFID systems, influenced by chip characteristics, is also related to the operation frequency. In real-world applications, RFID readings are often noisy, leading to potential failures in tag detection within the reader’s range due to weak or inconsistent signal readings.

Uncertainties in RFID applications can arise from various factors such as tag collisions, tag detuning (e.g., due to distance or the presence of metal/liquid affecting the field/absorption), tag misalignment, movement of tags, tag density, antenna design, cable type and length, mounting location, transponder or reader sensitivity, environmental factors, material density, operating frequency/coupling factor, etc. Missing data is one of the most significant ramifications of uncertainty. According to the Friis equation [36], the energy required for passive RFID sensor activation can diminish with distance, and this activation energy can be further reduced due to the multipath effect. In addition, regulations limit the output power of RFID readers. In scenarios with a single fixed RFID reader and passive sensor arrays, increasing the number of sensors in the far-field results in a larger radar cross section (RCS), which can lead to weak or absent signal readings as the measurement distance increases.

To enhance the sensing accuracy without changing system parameters like transmitter power and antenna direction gain, energy harvesting systems can be advantageous. Various energy harvesting systems are discussed in the literature, utilizing ambient and non-ambient energy sources. These systems, which include RF, piezoelectric, thermometric, photovoltaic, dynamic, and acoustic methods, convert collected energy into electrical energy and store it in batteries [37]. Typically, energy collectors in these systems are arrayed to capture a significant amount of power from the ambient. However, physical space constraints due to array size can limit the system efficiency.

Metamaterials and metasurfaces, or frequency-selective surfaces (FSS), are often used to address these limitations [38]. Metamaterials are artificial structural elements that offer advanced electromagnetic properties, consisting of either dielectric–dielectric or metal–dielectric materials. In dielectric–dielectric materials, each material’s dielectric constant varies, contributing to the working principle of these advanced materials.

Metamaterials are crafted by organizing small scatterers or holes in a consistent pattern across a spatial area to produce a specific bulk electromagnetic behavior. These properties are often unique, like negative refractive indices or near-zero indices, which are not typically found in natural materials. By arranging electrically tiny scatterers or holes in a two-dimensional pattern on a surface or interface, it is possible to obtain three-dimensional metamaterials. The surface variant is known as a metasurface.

Metasurfaces, as alternatives to metamaterials, have a wide range of applications. These include controllable innovative surfaces, miniaturized cavity resonators, waveguiding structures, absorbers, biomedical devices, and fluid tunable frequency-agile materials [39]. Metamaterial FSSs, such as artificial magnetic conductors and electromagnetic bandgaps, are used to create reactive impedance surfaces. Unlike fixed frequency metamaterials, whose frequency response is dictated by the static shape and spacing of unit cells, FSSs provide reflection properties with specific phases and amplitudes, allowing for frequency shifts in a single medium without being confined to a set frequency response.

FSSs have numerous benefits when used as a superstrate for various antennas. They are simpler to manufacture with microstrip technology compared to dielectric-type superstrates and require less thickness. Additionally, FSSs can suppress grating lobes, leading to the creation of high-gain antennas [40]. This versatility and utility make FSSs an attractive option in the development of advanced antenna systems and other electromagnetic applications.

Metamaterials and metasurfaces can be manufactured using various AM techniques, such as material extrusion, inkjet printing, stereolithography, multiphoton polymerization, laser chemical vapor deposition, aerosol jet printing, powder bed fusion, and laser rapid prototyping. One of the most popular AM methods for creating metamaterials is fused deposition modeling (FDM). FDM’s main advantages are the availability of materials and the capability to use multiple materials during the production process. However, producing custom composite filaments for metamaterials can be time-consuming, and the final product’s resolution is limited by the nozzle size [41].

In RFID systems, meta-surface-based RF absorbers are often used to mitigate undesirable multipath effects [42]. These absorbers consist of two-dimensional subwavelength periodic unit cells that align the incoming wave impedance with the absorber’s resonance frequency. Traditional methods involve radar-absorbing materials like polyurethane composite foams, ferrite, and carbon, but these can be bulky, heavy, and less cost-effective compared to meta-surface solutions [43].

Designing metamaterial or metasurface-based energy harvesting systems is complex, and the geometry must be produced smoothly as per the multi-material and multi-process steps. Challenges include scalability, resolution, uniformity, controllability, and mass production [44]. Some luminescent materials, like zinc sulfide with certain impurities, exhibit significant changes in dielectric constant when activated by UV light or other appropriate radiation, particularly those showing photoconductivity during luminescence. Changes in the dielectric constant of such materials can be up to 75%, with considerable dielectric loss even at moderate excitation intensities [45].

Phosphorescent materials, commonly referred to as glow-in-the-dark, possess the unique ability to absorb electromagnetic energy and gradually re-emit it as visible light. This process is scientifically termed luminance. Luminescence, broadly, is the capability of materials to emit light without an external light source. It is categorized based on the mode of excitation and includes various forms like fluorescence, phosphorescence, and delayed fluorescence. Other types of luminescence are chemiluminescence, radioluminescence, bioluminescence, electroluminescence, cathodoluminescence, sonoluminescence, triboluminescence, and thermoluminescence [46]. Photoluminescence, a specific category, involves light emission triggered by direct photoexcitation of the emitting materials and manifests in two forms, i.e., fluorescence and phosphorescence. This phenomenon is distinct from incandescence, as it does not necessitate high temperatures and typically does not produce noticeable heat. Photoluminescence occurs when materials absorb and store energy by exciting electrons through external light photons, subsequently releasing this stored energy as visible light.

### 3.10. Test Procedure with New Encapsulation Method

Fluorescence and phosphorescence are both phenomena related to a substance’s ability to absorb light energy and re-emit it at a longer wavelength and lower energy. The primary distinction lies in the duration of the light emission. Fluorescence emits light almost instantaneously and is observable only while the light source is active. In contrast, phosphorescent materials have the capacity to store absorbed light energy and release it over time, creating an afterglow that persists even after the exogenous light source is turned off. This glow can last from a few seconds to several hours, depending on the material’s composition. Due to this extended emission period, we use a phosphorescent material in our tests.

The tests involve using a phosphorescent material on both the RFID reader antenna and the RFID tags/sensors. We use a commercially available self-adhesive tape with a thickness of 5 mm, which contains a phosphorescent material that recharges with exposure to overhead or natural light. The first test involves a 3D-printed PP-encapsulated RFID sensor array, covered with the phosphorescent material (as shown in Figure 6). This sensor array is placed 3 λ away from the RFID reader antenna. Initially, the array is positioned in front of the phosphorescent material, and measurements are taken before and after its stimulation by light. Tests are first conducted without light stimulation, then in darkness, with the entire phosphorescent-covered side stimulated by circularly polarized light, and finally, with the entire phosphorescent-covered side excited by a light array matrix. All measurements are carried out within the relaxation time of the phosphorescent material.

In the second test procedure, the front of the RFID reader antenna is covered with self-adhesive tape containing phosphorescent material, and only one sensor is used. This sensor is then moved horizontally 43 cm to the left and then 43 cm to the right, using the utilized sliding door mechanism. RSSI data are collected for each 1 cm movement. The setup involves the RFID reader antenna placed 3 λ away from the sensor and measurements taken under various conditions: first without light stimulation, then in darkness with the entire phosphorescent-covered side stimulated by light, and finally, with the phosphorescent-covered side excited using a light array matrix. All measurements are carried out within the phosphorescent material’s relaxation time. The results from both tests mutually confirmed each other due to the two-way communication of the RFID system (as illustrated in Figure 7).

Based on the test results, when the phosphorescent material, which is made of partially crystalline and non-polar materials like PP, is placed in front of the sensor or antenna:When the entire surface is stimulated with light, the RFID signal level is attenuated in the dark due to an increase in the phosphorescent material’s dielectric constant.When the surface is exposed to a light array and specific boundary conditions are met, there is an observed increase in the RFID signal level. This enhancement in signal strength can be attributed to two key factors: changes in the dielectric constant of the phosphorescent material and the occurrence of Mie scattering during its relaxation time in the dark. These phenomena facilitate the harvesting of energy by the RFID sensor, which in turn leads to enhanced sensitivity and accuracy.

When the phosphorescent material is positioned behind the sensor or antenna:When the whole surface is stimulated with light, there is no change in the RFID signal level. In this scenario, the phosphorescent material acts as a reflector due to the increase in its dielectric constant.When the surface is stimulated with a light array, under specific boundary conditions, an increase in the RFID signal level is observed. This is again due to changes in the phosphorescent material’s dielectric constant and Mie scattering, with energy harvesting by the RFID sensor leading to improved performance. In these tests, an increase of approximately 2 dB in RSSI level is observed.

In addition, the measurements show that the half-power beam width (HPBW) of the RFID reader antenna before the stimulation (HPBW-1 in Figure 8) is significantly larger than the HPBW after the stimulation (HPBW-2 in Figure 8), indicating a notable beam narrowing effect. Furthermore, in the dark conditions post-stimulation, there is a noticeable increase in the RSSI levels. This suggests that the phosphorescent material’s interaction with the light array alters the RFID reader antenna’s radiation pattern and enhances signal reception.

In materials with prominent electrical or magnetic properties, the real component of the average power over a period is linked to the imaginary parts of permittivity (ε = ε′ − jε′′) and permeability (μ = μ′ − jμ′′), as described by the Poynting theorem. This indicates that any resonance observed in a passive polarizable array or metasurface is not just a passive response but can serve as an indicator of energy storage within the material [47]. This principle highlights the capacity of these materials to store and potentially release energy under certain conditions, a phenomenon of considerable importance in various applications involving electromagnetic fields and waves.

At frequencies higher than microwave, particularly in the infrared and visible bands, engineering duly thin absorbers that produce equally strong electric and magnetic induced currents poses a significant challenge. This difficulty primarily arises from the scarcity of naturally magnetic materials suitable for optical applications [44]. However, in the VHF and UHF bands, issues like narrowband characteristics and low antenna efficiency can be addressed using magneto-dielectric materials. These materials not only enhance antenna performance but also contribute to reducing the antenna size and RCS [48].

In this work, we leverage an energy-storing material to boost RSSI levels and mitigate the multipath effect, thereby reducing instances of missing data in RFID systems. Rather than utilizing metamaterials for antenna design, we employ phosphorescent material within the Mie resonance field. This approach is particularly relevant in electromagnetic (EM) wave theory, where the magnetic field plays a crucial role, especially in near-field communication scenarios within the first Fresnel zone. By utilizing phosphorescent materials in this context, we can enhance the signal strength and reliability, addressing common challenges in RFID communication systems.

When phosphorescent material is placed in front of the RFID sensor array, a noticeable signal attenuation is observed after the entire surface is exposed to light, with subsequent RSSI measurements conducted in a dark environment. This attenuation is a result of increased conductivity, causing the surface to act as a high-impedance reflector. In contrast, when only a light array stimulates the surface covered with the phosphorescent material, an increase in signal level, approximately 2 dB, is detected. This enhancement in signal strength is further amplified through photonic cavities within the magnetic field, leading to an increase in signal levels by around 8 dB, contingent on the type of propagation, whether it is near-field or far-field. This occurs due to the absorption of energy and the creation of a frequency-selective surface.

In a direct-current field, partially crystalline polymers exhibit a unique behavior where charges accumulate at the interfaces between crystalline and amorphous regions. This phenomenon occurs due to the polarization effect, where each crystallite grain acts as a dipole, thereby becoming a primary source for trapping charges [6]. As a result of this charge accumulation, each radiating cavity within the polymer forms a conductive surface. Importantly, the conductivity of this surface is not static but changes over time. This dynamic nature of conductivity means that the dimensions of the conductive area, which are directly related to the resonance frequency, also vary over time. This variation in the conductive area and resonance frequency contributes to the unique properties of these materials when used in RFID systems.

Ultimately, the integration of our novel photoluminescence-based technique, which significantly enhances RFID signal levels, with classification algorithms yields a classification performance as high as 98%, effectively matching the accuracy obtained in scenarios with no data gap.

## 4. Concluding Remarks

Within the realm of SHM, the displacement of objects was accurately estimated using RFID passive sensors and ML algorithms, achieving high accuracy. However, increasing the number of sensors led to the challenge of missing data, which conventional techniques like interpolation and imputation were unable to fully address. To tackle this challenge, a unique strategy was implemented to augment the sensitivity of passive RFID sensors encapsulated in a semi-crystalline polymer. This enhancement was achieved using an innovative photoluminescence-based technique, specifically designed for RF signal amplification. Through energy absorption in photonic cavities within the magnetic field, RF levels were boosted by approximately 2 dB to 8 dB, depending on whether it was near-field or far-field propagation. This advancement effectively resolved the missing data issue in RFID communication without the need to alter system parameters like power or sensor number. This innovative approach allows for the formation of radiation patterns remotely through light, as opposed to traditional material shaping used in metamaterials. The introduction of this method marks a significant step forward and opens new possibilities for the development of advanced smart antennas for various applications including SHM.

The versatility of our innovative approach extends significantly into practical SHM applications. For instance, it can revolutionize the way critical infrastructure is monitored and maintained by facilitating the SHM of high-rise buildings or long-span bridges. The developed technology not only promises to enhance the accuracy and reliability of structural assessments but also introduces a proactive means of identifying potential issues before they escalate. By leveraging the enhanced sensitivity of passive RFID sensors within such structures, engineers and maintenance teams can obtain real-time, precise data on displacements or stresses, enabling timely interventions and ensuring the longevity and safety of these engineering marvels.

Lastly, we presented our interpretation, grounded in our present knowledge, of the mechanism by which photoluminescent material contributes to minimizing data gaps. While we have endeavored to provide a comprehensive account, we acknowledge that our explanation may not exhaustively cover the intricacies of this unexpected phenomenon. This acknowledgment stems from our recognition of the complexity and novelty of the interaction between photoluminescent materials and RFID signal enhancement. Consequently, we consider the detailed exploration of this phenomenon (a blend of serendipitous discovery and scientific inquiry) as a promising direction for future research. We believe that subsequent studies ought to delve both theoretically and empirically into this subject, aiming to unravel the underlying principles and potential applications further. This future work can significantly advance our understanding and lead to further novel approaches for addressing data gaps in RFID-based systems as well as opening new avenues for the application of photoluminescent materials in various technological domains.

## Figures and Tables

**Figure 1 sensors-24-01233-f001:**
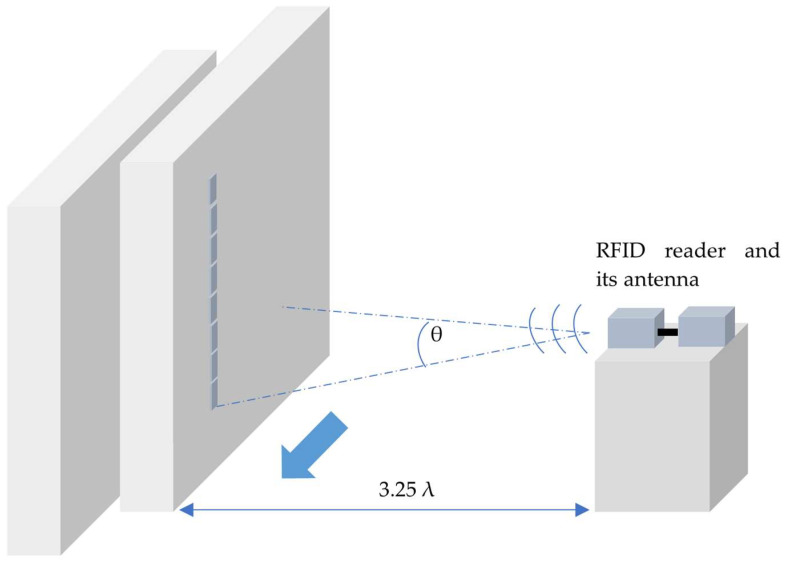
Test platform for estimation of DOA.

**Figure 2 sensors-24-01233-f002:**
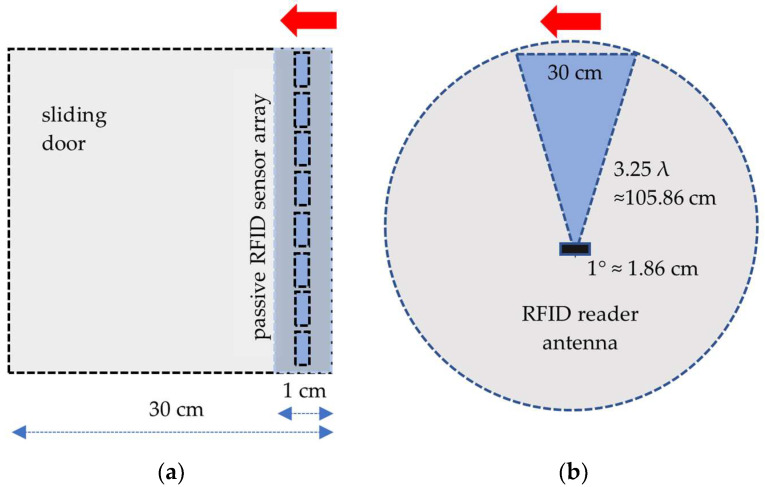
Sweeping sensor arrays and test platform for estimation of DOA (**a**): sensor array front view, (**b**): measurement top view.

**Figure 3 sensors-24-01233-f003:**
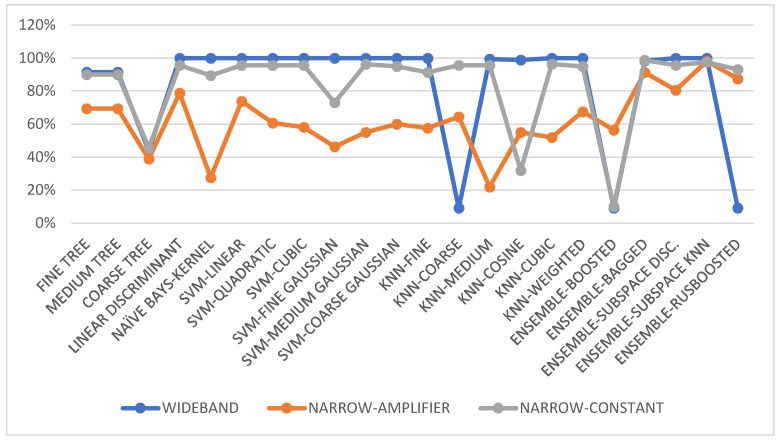
Classification performance in terms of accuracy (%) with the augmented dataset for Test-1: wideband, Test-2: narrowband with an amplifier, and Test-3: narrowband with constant gain.

**Figure 4 sensors-24-01233-f004:**
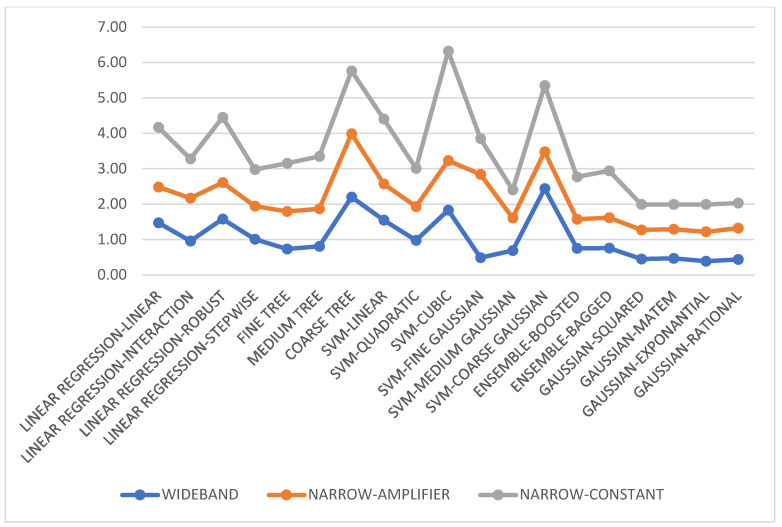
Regression performance in terms of root mean square error (RMSE) with the augmented dataset for Test-1: wideband, Test-2: narrowband with an amplifier, and Test-3: narrowband with constant gain.

**Figure 5 sensors-24-01233-f005:**
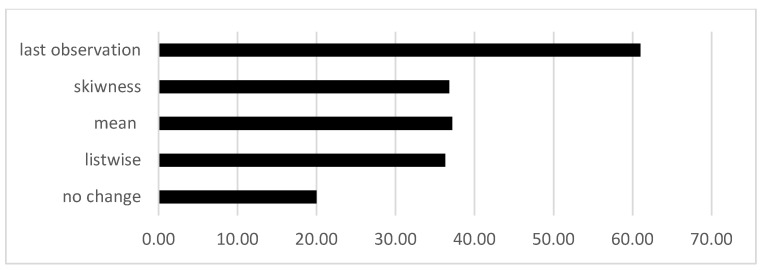
Classification performance improvement using deletion and imputation methods.

**Figure 6 sensors-24-01233-f006:**
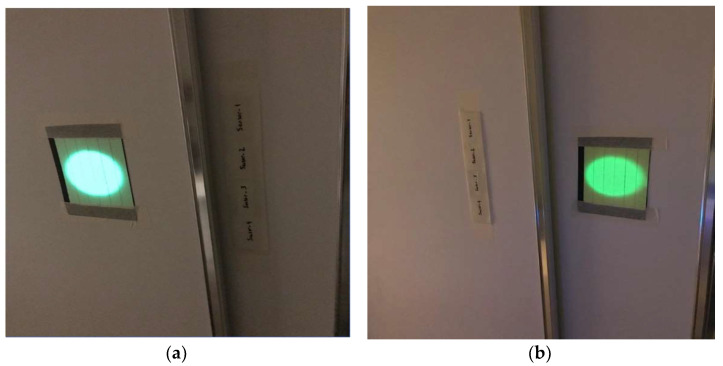
The phosphorescent material (on self-adhesive tape with a thickness of 5 mm) is first put in front of the sensor array (**a**); then, is installed on the back (**b**).

**Figure 7 sensors-24-01233-f007:**
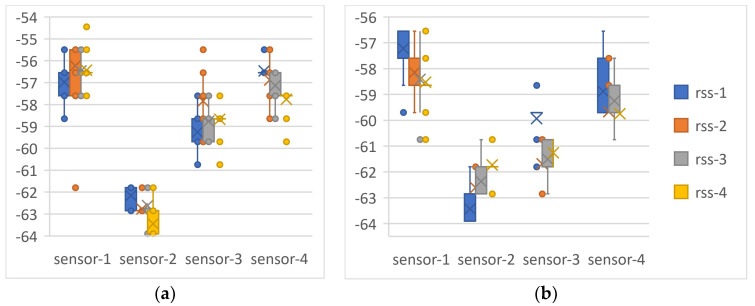
Results of the first test procedure. The phosphorescent material is first put in front of the sensor array (**a**), then installed at the back (**b**). rss-1 (dBm): no phosphorescent material; rss-2 (dBm): with phosphorescent material but no excitation by light; rss-3 (dBm): with phosphorescent material and light excitation by circular light; rss-4 (dBm): with phosphorescent material and excitation by a light matrix.

**Figure 8 sensors-24-01233-f008:**
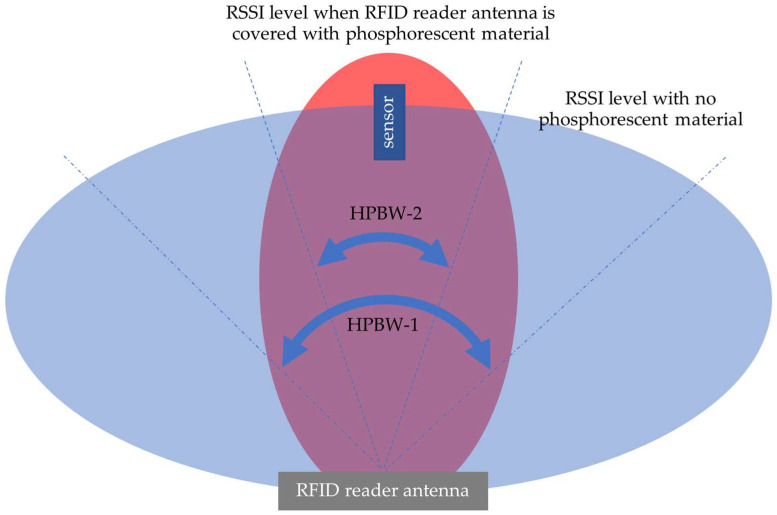
Results of the second test procedure. The front of the RFID reader antenna, situated 3 λ away from the sensor, is covered with the phosphorescent material, which is then stimulated with a light array. The results show HPBW-1 >> HPBW-2 with RSSI levels increased.

## Data Availability

Data are contained within the article.

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
