# Peer review of "Displacement Estimation via 3D-Printed RFID Sensors for Structural Health Monitoring: Leveraging Machine Learning and Photoluminescence to Overcome Data Gaps"

_sensors, 2024, doi:10.3390/s24041233_

Round 1

Reviewer 1 Report

Comments and Suggestions for Authors This paper proposed to enhance the sensitivity of 3D-printed passive RFID sensor arrays using 16 a novel photoluminescence-based RF signal enhancement technique. After reading this paper, following comments should be addressed: (1)  we is not suggested to  be used in the paper according to the scientific writing rules. (2) what is the boundary of the untra high frequency. (3)pls prospect the use of this technlogy in high-rise buildings or lang-span bridges.

Author Response

We thank the reviewer for their time and effort in reviewing our manuscript and providing insightful comments. Below, we respond to each comment, presented in italic font. We have highlighted the changes made to address these comments in green in the revised manuscript.

This paper proposed to enhance the sensitivity of 3D-printed passive RFID sensor arrays using a novel photoluminescence-based RF signal enhancement technique. After reading this paper, following comments should be addressed:

(1) we is not suggested to be used in the paper according to the scientific writing rules.

Response: We are not aware of any scientific writing rule prohibiting the use of active language. Nonetheless, we have applied it judiciously in the revised manuscript.

(2) what is the boundary of the untra high frequency.

To address this comment, we added the following sentence as a footnote to page 3 of the revised manuscript:

“Ultra-high frequency refers to the radio frequency range between 300 MHz and 3 GHz.”

(3)pls prospect the use of this technlogy in high-rise buildings or lang-span bridges.

To address this comment, we added the following sentences to section 4 of the revised manuscript:

“The versatility of our innovative approach extends significantly into practical SHM applications. For instance, it can revolutionize the way critical infrastructure is monitored and maintained by facilitating the SHM of high-rise buildings or long-span bridges. The developed technology not only promises to enhance the accuracy and reliability of structural assessments but also introduces a proactive means of identifying potential issues before they escalate. By leveraging the enhanced sensitivity of passive RFID sensors within such structures, engineers and maintenance teams can obtain real-time, precise data on displacements or stresses, enabling timely interventions and ensuring the longevity and safety of these engineering marvels.”

Reviewer 2 Report

Comments and Suggestions for Authors

A review of

Displacement estimation via 3D-printed RFID sensors for structural health monitoring: Leveraging machine learning and photoluminescence to overcome data gaps

By Metin Pekgor et al

This paper introduced machine learning (ML) and photoluminescence to address the data gaps that frequently occurs during structural health monitoring (SHM), especially with the RFID sensors. Data gaps or data missing or Null data has been a common issue in the data acquisition process of SHM. The proposed ML based technique can be applied to other sensors, not only the RFID sensors. To this end, this paper can be beneficial to the filed of structural health monitoring. However, following issues shall still be addressed before publication:

One major issue is that the mechanism of RFID is not explained in the section of “Meterials and Methods”. It is hard to follow how the displacement of the sliding door is obtained from the recorded data? What type of data is measured? How is this data transmitted to the position of the host structure?

Most of the figures in this paper are badly prepared, especially Figure 3. They are not clearly enough for an academic paper.

Why photoluminescence material can assist to reduce data gaps? The mechanism shall be explored and justified. I believe this is a great novelty. However, this is not clearly explained in the current version.

Author Response

We thank the reviewer for their time and effort in reviewing our manuscript and providing insightful comments. Below, we respond to each comment, presented in italic font. We have highlighted the changes made to address these comments in green in the revised manuscript.

This paper introduced machine learning (ML) and photoluminescence to address the data gaps that frequently occurs during structural health monitoring (SHM), especially with the RFID sensors. Data gaps or data missing or Null data has been a common issue in the data acquisition process of SHM. The proposed ML based technique can be applied to other sensors, not only the RFID sensors. To this end, this paper can be beneficial to the filed of structural health monitoring. However, following issues shall still be addressed before publication:

One major issue is that the mechanism of RFID is not explained in the section of “Meterials and Methods”. It is hard to follow how the displacement of the sliding door is obtained from the recorded data? What type of data is measured? How is this data transmitted to the position of the host structure?

At the beginning of section 3, we refer readers to our previous publication [30] for an in-depth exploration of the related signal modeling, underlying assumptions, fundamental principles of RFID sensing, and data processing pipeline. This decision was made to prevent redundancy and to maintain the conciseness of the current manuscript. Furthermore, within section 3, we elaborate on the nature of the data collected (particularly sections 3.3 and 3.5), alongside the methodology employed to estimate the direction of arrival and displacement values from the collected data. While this may not be immediately apparent, we assure readers that a thorough examination will reveal the detailed integration of these elements. Additionally, to foster transparency and facilitate further research, we plan to make the data and code used to achieve the findings reported in this manuscript publicly accessible, enabling interested readers to replicate our results and gain additional understanding.

Moreover, to further address this comment, we added the following sentences to the end of section 3.5 of the revised manuscript.

“The data collected by the RFID reader is sent to a local computing device for analysis. The method of data transmission and the communication channel used can differ based on the specific requirements of the application.”

[30] M. Pekgor, et al., “Displacement estimation using 3D-printed RFID arrays for structural health monitoring,” Sensors, 2022. 22, DOI: 10.3390/s22228811.

Most of the figures in this paper are badly prepared, especially Figure 3. They are not clearly enough for an academic paper.

In the revised manuscript, we have replaced all figures with their high-resolution versions and removed the previous Figure 3 due to its redundancy.

Why photoluminescence material can assist to reduce data gaps? The mechanism shall be explored and justified. I believe this is a great novelty. However, this is not clearly explained in the current version.

In Section 3.10, we have detailed our current understanding of how the utilized photoluminescence material contributes to minimizing data gaps, to the best of our knowledge. We recognize that this explanation does not fully elucidate the serendipitous nature of this phenomenon and view it as an area for future research, inviting further theoretical and empirical investigation. Therefore, to address the comment, we added the following paragraph to section 4 of the revised manuscript.

“Lastly, we presented our interpretation, grounded in our present knowledge, of the mechanism by which photoluminescence material contributes to minimizing data gaps. While we have endeavored to provide a comprehensive account, we acknowledge that our explanation may not exhaustively cover the intricacies of this unexpected phenomenon. This acknowledgment stems from our recognition of the complexity and novelty of the interaction between photoluminescence materials and RFID signal enhancement. Consequently, we consider the detailed exploration of this phenomenon (a blend of serendipitous discovery and scientific inquiry) as a promising direction for future research. We believe that subsequent studies ought to delve both theoretically and empirically into this subject, aiming to unravel the underlying principles and potential applications further. This future work can significantly advance our understanding and lead to further novel approaches for addressing data gaps in RFID-based systems as well as opening new avenues for the application of photoluminescence materials in various technological domains.”

Reviewer 3 Report

Comments and Suggestions for Authors

1. It is suggested that the review of RFID positioning systems should be improved because the logic supporting the research is not clearly illustrated. As there are many classification methods, the reader could get confused without a clear structure.

2. For part 3, it could be clearer to add a paragraph to introduce the overall research structure before the classical method. 

3. Resolution and aspect ratio is not correct for figure 10.

4. The theory background for improving the sensitivity of RFID tag with photoluminescent material is explained in simple sentences rather than strong theoretic calculations and could be improved.

Author Response

We thank the reviewer for their time and effort in reviewing our manuscript and providing insightful comments. Below, we respond to each comment, presented in italic font. We have highlighted the changes made to address these comments in green in the revised manuscript.

  1. It is suggested that the review of RFID positioning systems should be improved because the logic supporting the research is not clearly illustrated. As there are many classification methods, the reader could get confused without a clear structure.

We understand there are concerns regarding the clarity of the logic supporting our research and the structure of our classification of RFID positioning systems. In addressing this, we wish to clarify that the foundational logic of our research is primarily detailed in the Introduction section. This segment aims to set the stage for our study, distinguishing it from the Review of RFID Positioning Systems section, which focuses on providing a comprehensive overview of related literature rather than exploring our research logic in depth.

Our classification of existing work adheres closely to the methodologies and categorizations presented in the referenced publications. We aimed to encompass a broad spectrum of the field to ensure a thorough review, which might have contributed to the perception of complexity and confusion. Recognizing this potential for misunderstanding, we have diligently reviewed and revised our material to enhance clarity and coherence, striving for a balance between comprehensiveness and readability.

We are committed to improving our work continually and acknowledge that despite our revisions, there may still be areas that could benefit from further clarification. We welcome more detailed feedback on specific aspects that may still be confusing, allowing us to refine our review further.

  1. For part 3, it could be clearer to add a paragraph to introduce the overall research structure before the classical method.

To address this comment, we added the following sentences to the first paragraph of section 3.

“Here, the objective is to explore the development of new methods for displacement estimation in SHM utilizing 3D-printed RFID sensors. Our methodology involves the estimation of DOA values through the application of ML algorithms, embracing both supervised learning paradigms of classification and regression. Given the data-driven nature of our approach, which necessitates high-quality data, we introduce a novel method to enhance RFID signal levels through exploiting the synergistic impact of photoluminescence on RF signals and antenna radiation patterns. The proposed technique helps address potential data gaps and improve the reliability and accuracy of ML-based displacement estimation in SHM applications.”

  1. Resolution and aspect ratio is not correct for figure 10.

In the revised manuscript, we have replaced all figures with their high-quality version for enhanced clarity. We have also removed the redundant Figure 10 to streamline the presentation.

  1. The theory background for improving the sensitivity of RFID tag with photoluminescent material is explained in simple sentences rather than strong theoretic calculations and could be improved.

In Section 3.10, we have detailed our current understanding of how the utilized photoluminescence material contributes to minimizing data gaps, to the best of our knowledge. We recognize that this explanation does not fully elucidate the serendipitous nature of this phenomenon and view it as an area for future research, inviting further theoretical and empirical investigation. Therefore, to address the comment, we added the following paragraph to section 4 of the revised manuscript.

“Lastly, we presented our interpretation, grounded in our present knowledge, of the mechanism by which photoluminescence material contributes to minimizing data gaps. While we have endeavored to provide a comprehensive account, we acknowledge that our explanation may not exhaustively cover the intricacies of this unexpected phenomenon. This acknowledgment stems from our recognition of the complexity and novelty of the interaction between photoluminescence materials and RFID signal enhancement. Consequently, we consider the detailed exploration of this phenomenon (a blend of serendipitous discovery and scientific inquiry) as a promising direction for future research. We believe that subsequent studies ought to delve both theoretically and empirically into this subject, aiming to unravel the underlying principles and potential applications further. This future work can significantly advance our understanding and lead to further novel approaches for addressing data gaps in RFID-based systems as well as opening new avenues for the application of photoluminescence materials in various technological domains.”

Round 2

Reviewer 2 Report

Comments and Suggestions for Authors

issues have been addressed.